# De Novo Prediction of Drug Targets and Candidates by Chemical Similarity-Guided Network-Based Inference

**DOI:** 10.3390/ijms23179666

**Published:** 2022-08-26

**Authors:** Carlos Vigil-Vásquez, Andreas Schüller

**Affiliations:** 1Department of Molecular Genetics and Microbiology, School of Biological Sciences, Pontificia Universidad Católica de Chile, Santiago 8331150, Chile; 2Institute for Biological and Medical Engineering, Schools of Engineering, Medicine and Biological Sciences, Pontificia Universidad Católica de Chile, Santiago 7820436, Chile

**Keywords:** target prediction, drug–target interaction, network-based inference, drug repositioning, drug discovery

## Abstract

Identifying drug–target interactions is a crucial step in discovering novel drugs and for drug repositioning. Network-based methods have shown great potential thanks to the straightforward integration of information from different sources and the possibility of extracting novel information from the graph topology. However, despite recent advances, there is still an urgent need for efficient and robust prediction methods. Here, we present SimSpread, a novel method that combines network-based inference with chemical similarity. This method employs a tripartite drug–drug–target network constructed from protein–ligand interaction annotations and drug–drug chemical similarity on which a resource-spreading algorithm predicts potential biological targets for both known or failed drugs and novel compounds. We describe small molecules as vectors of similarity indices to other compounds, thereby providing a flexible means to explore diverse molecular representations. We show that our proposed method achieves high prediction performance through multiple cross-validation and time-split validation procedures over a series of datasets. In addition, we demonstrate that our method performed a balanced exploration of both chemical ligand space (scaffold hopping) and biological target space (target hopping). Our results suggest robust and balanced performance, and our method may be useful for predicting drug targets, virtual screening, and drug repositioning.

## 1. Introduction

Traditionally, drug design followed the paradigm of “one drug, one target” seeking a “magic bullet” drug against a single molecular target, assuming that a single target was responsible for a pathology [1,2]. In recent years, this paradigm is slowly shifting toward the awareness that a single compound may act on several targets simultaneously, a concept coined polypharmacology [3]. Many drugs bind multiple targets, either intended (polypharmacology) or not intended (off-targets), and this is especially true in complex diseases such as cancer and central nervous system diseases [4]. It was shown that in recent years new drugs increasingly interfere with multiple targets [5]. Identifying all potential targets of a small molecule is a goal of polypharmacology drug design. Identifying a compound’s target profile is also important to flag off-targets, find new targets of known drugs (drug repurposing), and to deorphanize ligands without known targets and targets without known ligands. Experimental profiling of drug candidates against a large panel of diverse targets is cost and time intensive. Undetected off-targets may result in adverse drug effects and may cause failures in clinical trials [6,7]. Desired secondary targets may contribute to improved efficacy of new drugs [3]. Several computational methods for target identification and predicting drug–target interactions (DTIs) have been proposed to bridge the gap between ligand and target spaces [8]. Linear models are based on chemical information of the ligands [9,10,11], structural information of the targets [12,13], and combined information [14]. Machine learning methods for target prediction include support vector machines (SVM) [15,16], Gaussian kernels [17], nearest neighbor classifier [18], Bayesian models [19], Boltzmann machines [20], and more recently, deep learning methods [21,22,23]. The general idea of most of these methods is the similarity property principle: similar compounds are likely to have similar bioactivities [24], which can be extrapolated to proteins: targets with a similar structure are likely to have similar functions. Despite these successes, the prediction of targets for drugs remains challenging. Network-based methods provide an alternative approach and have been employed successfully. Bipartite graph learning [25], CSNAP and 3DCSNAP [26,27], nAnnolyze [28], and DASPfind [29] serve as examples of different network approaches used for target prediction, employing different graph theory-centric concepts, i.e., topology, centrality, shortest paths, and random paths, respectively. Recently, continuous interest has been put in methods based on the network-based inference (NBI) formalism. NBI, proposed by Zhou et al. [30] and adapted for the prediction of DTIs by Cheng et al. [31], is based on a resource-spreading algorithm within a bipartite drug–target network, which distributes a quantifiable measure throughout the network to reach a target node, i.e., a biological target. This method demonstrated high predictive performance; however, it cannot predict targets for compounds not connected to the initial network (a.k.a. de novo prediction). Wu et al. [32] proposed Substructure-Drug-Target Network-based inference (SDTNBI) to solve this limitation, which incorporates chemical substructures into the bipartite network enabling the prediction of targets for novel chemical entities with improved performance. However, their method is limited to binary molecular fingerprints, preventing the utilization of other types of molecular descriptions that are useful for different tasks in drug development, e.g., scaffold hopping [33,34,35].

Here, we propose SimSpread, a new method for the prediction of drug–target interactions which employs a tripartite drug–drug–target network where resources are distributed based on the NBI formalism. The first network layer (drugs) can be understood as a feature layer by which small molecules are described as a vector of similarity indices to other compounds (Figure 1A). This feature layer, which consists of drugs with known target annotations and bioactivities, is connected to the second layer by means of chemical similarity (Figure 1B). An edge is formed between the first and second layers if the chemical similarity is above an adjustable threshold. The third layer corresponds to the target annotations. Here, an edge is formed to a drug of the second layer if the annotated bioactivity value is below a concentration threshold (Figure 1B). Target predictions are performed by applying a resource-spreading algorithm on the tripartite network (Figure 1C,D). Here, we first describe the optimization of free method parameters with multiple cross-validation procedures over a series of benchmark datasets. Next, we show that SimSpread’s performance was competitive with or exceeded other network-based and chemical similarity-based algorithms, and was robust in external time-split validation. Finally, we show that SimSpread demonstrated a balanced exploration behavior of chemical and biological space, enabling scaffold-hopping and covering diverse targets.

## 2. Results and Discusion

We optimized free method parameters on five benchmark datasets previously employed in similar works (Enzyme, Ion Channel, GPCR, Nuclear Receptor) [25] where the number of DTIs ranges from 90 to 2926, and the larger Global dataset with 10,185 DTIs [32]. Parameter optimization was performed with leave-one-out (LOO) and 10-times 10-fold cross-validation (CV). In addition, we validated the robustness of the predictions on an external time-split dataset derived from ChEMBL. Please refer to the Section 3 for additional details.

### 2.1. Selection of Optimal Method Parameters

Free method parameters included (i) the similarity cutoff α to link a query molecule to the first layer and to connect the first layer to the second layer, (ii) the choice of molecular descriptor, and (iii) the weighting scheme of the resources assigned to the first layer. We systematically optimized the similarity cutoff α ranging from 0 to 1 with a step size of 0.05 for bit-based molecular descriptors and a range from 0.8 to 1.0 with a step size of 0.01 for the Mold ${}^{2}$ molecular descriptor, which has a distribution of Tanimoto coefficients shifted to higher values (Appendix A). Prediction results based on LOOCV indicate that the best performance was achieved with the circular fingerprints ECFP4 and FCFP4, with optimal α values ranging from 0.2 to 0.3 (Figure 2 and Appendix A). We implemented two different weighting schemes. In the binary variant SimSpread ${}_{bin}$, an initial resource value of “1” is assigned to nodes with a Tanimoto coefficient greater or equal to α, or ”0” otherwise. In the similarity-weighted variant SimSpread ${}_{sim}$, the assigned resource value corresponds to the actual value of the Tanimoto coefficient for similar compounds, or zero otherwise. In LOOCV, SimSpread ${}_{sim}$ performed slightly better (2.1% on average) than SimSpread ${}_{bin}$ when comparing performance using the optimal cutoff. For 10-times 10-fold CV, we observed a similar behavior, achieving the best predictive performance using circular fingerprints ECFP4 and FCFP4 with optimal α values ranging from 0.2 to 0.4, where SimSpread ${}_{sim}$ performed 7.2% better on average than SimSpread ${}_{bin}$ (Appendix A).

The first network layer is effectively an abstracted feature layer where compounds are described by their similarity to other compounds. This permits the added flexibility of freely choosing any type of molecular descriptor. We included the real-valued Mold ${}^{2}$ descriptor in our analyses, which consists of a diverse set of 777 individual one-dimensional and two-dimensional molecular descriptors, such as atoms counts, physicochemical properties, and topological features calculated from the two-dimensional chemical structure [34]. Because of the distribution of similarity values, this descriptor required higher α values. Here, the continuous formula of the Tanimoto metric was employed [36]. However, overall performance was inferior to the circular fingerprints (Figure 2, Appendix A). Not surprisingly, the distributions of AuPRC vs. Tanimoto cutoff (Figure 2) resembled the Tanimoto coefficient frequency distributions of each descriptor (Appendix A). Therefore, we advise optimizing α should a new descriptor (not tested in this study) be chosen.

Based on this analysis, we settled on the following optimized parameters for robust performance on all five benchmark datasets: ECFP4 descriptors with α=0.2 and SimSpread ${}_{sim}$ weighting.

### 2.2. Prediction Performance Compared to Other Methods

We compared the prediction performance of SimSpread to the related SDTNBI method, in which the feature layer corresponds to a binary molecular fingerprint [32], and a classical *k*-nearest neighbor approach (*k*-NN). For the latter *k* = 1 was chosen, based on previous analyses [9,10,11]. Performance was estimated by LOO and 10-times 10-fold CV on all five benchmark datasets for the ECFP4 descriptor and Tanimoto coefficient as similarity measure. For SimSpread variants, the similarity cutoff corresponds to the one that maximizes the mean AuPRC for each dataset. Our results suggest that SimSpread ${}_{sim}$ outperformed SimSpread ${}_{bin}$ and 1-NN in all cases, based on AuPRC (Figure 3). In most cases, SimSpread ${}_{sim}$ further obtained better results than SDTNBI in 7 out of 10 comparisons, 9.8% improvement in AuPRC performance (Appendix A). For median early-recognition performance, SimSpread variants achieved the best performance in 23 of 30 (76.7%) comparisons (Appendix A). Finally, median binary prediction performance shows that SimSpread variants achieved the best performance in 19 of 30 (66.3%) comparisons (Appendix A). Average performance for overall, early-recognition and binary prediction metrics can be found in Appendix A, respectively, for LOOCV, and Appendix A, respectively, for 10-times 10-fold CV.

For completeness, we included the original NBI algorithm in this comparative analysis. Since NBI cannot predict targets for novel compounds not included in the network, we modified the 10-times 10-fold CV scheme to consider DTIs instead of drugs in the hold-out. This analysis indicated that our method consistently performed better than NBI (Appendix A).

We conclude that SimSpread is competitive with other network-based (SDTNBI) and chemical similarity-based (*k*-NN) methods and, in many cases, surpassed their prediction performance.

### 2.3. Prediction Performance Estimated by External Time-Split Validation

It is known that cross-validation tends to overestimate the true predictive power of computational methods [37]. Hence, we evaluated the performance of SimSpread on an external time-split dataset derived from ChEMBL [38]. In time-split validation, the training set corresponds to information available up to a specific point in time, while the test set corresponds to all posterior or newer information. This type of validation was suggested to provide a more realistic testing scenario less prone to overestimation [37]. Here, the drug–target network was derived from ChEMBL version 24 (June 2018), and test compounds were derived from ChEMBL version 28 (February 2021). The validation task was to predict interactions between test compounds from ChEMBL28, which were published after June 2018, to training targets of ChEMBL24. Previously optimized parameters for SimSpread were employed (ECFP4 descriptors with α = 0.2), and the results were compared to SDTNBI and 1-NN.

As expected, our results indicate that the general prediction performance diminished substantially compared to our LOOCV (83.3% performance loss on average for all four methods from 0.60 to 0.10 median AuPRC) and 10-times 10-fold CV (70.6% performance loss on average from 0.37 to 0.10 median AuPRC) of the previous section (Figure 4, Table 1). The median performance of SimSpread ${}_{sim}$ was superior to SimSpread ${}_{bin}$ (by 8.43% ± 8.49% on average) and superior to SDTNBI (by 33.52% ± 24.99% on average) considering all evaluation metrics employed (Table 1 and Appendix A). Interestingly, the classical 1-NN approach outperformed all network-based methods in the statistics analyzed by an average of 9.47% ± 8.53%. (Figure 4, Table 1 and Appendix A). Median and average performance for overall, early-recognition and binary prediction metrics for time-split cross-validation can be found in Appendix A throughout Appendix A.

We were interested in a possible dependence of the prediction performance on the target degree, i.e, the number of annotated ligands per target. We performed linear regression analyses of all performance metrics of the time-split validation vs. target degree (Appendix A). Of eight performance metrics analyzed, only three depended significantly on target degree (*p* < 0.05, Wald test with t distribution). Effect sizes (coefficients of determination) were low (0.00004 < R ${}^{2}$ < 0.1). We conclude that in our time-split dataset performance did not generally depend on the number of annotated ligands per target. We believe that our network-based SimSpread method is particularly suited for prediction in datasets with low target degrees.

We consider that this external time-split validation is a more realistic approach to estimate prediction performance in a real-life scenario. The low median AuPRC (0.10 on average, Table 1) is alarming. However, likely a more relevant metric to estimate attrition rates for wet lab experiments is R@20, i.e., recall (percentage of retrieved annotated targets) of the top-20 predictions. In the 10-times 10-fold CV analysis of the previous section, all four methods performed similarly well (on average 75.9% ± 14.6% median R@20, Appendix A). In time-split validation this value dropped to 50.0% ± 13.6% on average (Table 1). Here, 1-NN (median R@20 = 66.7%) and SimSpread (median R@20 = 50.0%) demonstrated robust performance. This means that in 50% of all predictions, we can expect at least half of the true targets within the top-20 predictions. However, it should also be noted that these recall values per test ligand produced an interquartile range of 1.0, i.e., a maximally large dispersion. Thus, it is possible to retrieve all relevant targets within the top-20 predictions in a real-world prospective prediction scenario, or zero. This large dispersion is of concern and may increase attrition rates substantially. Developing more robust methods with less dispersion should be a goal of future research.

### 2.4. Analysis of the Exploration Behavior in Ligand and Target Space

In addition to the numerical performance, we were interested in the exploration behavior of the analyzed methods of both the chemical ligand space and the biological target space. All four methods produced result lists with different high-scoring drug–target interaction predictions, some of which were partially overlapping. Our goal was to understand the capacity to produce results for ligands with diverse and novel chemical scaffolds (scaffold hopping). Moreover, we were interested in predictions for many diverse targets (target hopping). As detailed in the Section 3, we sorted predicted DTIs of the time-split validation set by score and considered the top-250 predictions, a number we considered reasonable for informed manual inspection (cherry picking). Chemical space was represented by the 4096 bit-sized ECFP4 fingerprint of Murcko scaffolds [39], and target space was encoded by a binary vector (presence/absence) of 1185 gene ontology (GO) terms. We generated 2D representations of these spaces with the help of the UMAP algorithm (Figure 5) [40]. In these maps, gray dots represent all Murcko scaffolds or all targets present in the ChEMBL24→28 dataset, respectively. For the chemical space, blue and orange dots correspond to molecular scaffolds of the ligands of the top-250 high-scoring DTIs, where blue scaffolds were known, i.e., also found in the training set (ChEMBL24), and orange scaffolds were novel (exclusively found in the test set). As for target space, blue dots represent the biological targets of the top 250 DTIs.

Our 1-NN implementation generated high-scoring predictions for a small number of known (52) and novel (16) scaffolds (Figure 5A), while a large number of diverse targets (172) were retrieved (Figure 5E). In contrast, SDTNBI predicted high-scoring DTIs for a large number of known (125) and novel (27) scaffolds (Figure 5B) but a small number of targets (73) (Figure 5F). For SimSpread ${}_{bin}$, we found an intermediate number of known (86) and novel (19) molecular scaffolds in the top-250 high-scoring DTIs (Figure 5C), while the number of predicted targets (78) was similar in quantity to those found with SDTNBI (Figure 5G). Lastly, for SimSpread ${}_{sim}$, we found an intermediate number of known (92) and novel (20) molecular scaffolds and an intermediate number of predicted targets (100) (Figure 5D,H).

To analyze whether this balanced exploration behavior of SimSpread ${}_{sim}$ was dependent on the number of highest scoring DTIs considered in the analysis (i.e., the length of the sorted result list), we extracted result lists of increasing length *L* (from 10 to 1000 in steps of 10). Then, we plotted the number of unique scaffolds (known + novel) and unique targets present in the top *L* DTIs against the number of DTIs (*L*). We observed a similar trend as before. For molecular scaffold diversity, the trend was 1-NN < SimSpread ${}_{bin}$ = SimSpread ${}_{sim}$ < SDTNBI (Figure 5I), while it was the inverse for target diversity: SDTNBI < SimSpread ${}_{bin}$ < SimSpread ${}_{sim}$ < 1-NN (Figure 5J). Plotting the number of unique scaffolds against the number of unique targets further illustrates this balanced behavior of SimSpread in exploring chemical ligand and biological target space (Figure 5K). This balanced exploration behavior suggests that SimSpread is suitable for both scaffold and target hopping, and may be useful in virtual screening and target prediction tasks.

## 3. Materials and Methods

### 3.1. Dataset Selection and Preparation

Five benchmark datasets were employed (Table 2): (i) Nuclear Receptor, (ii) G Protein-Coupled Receptors, (iii) Ion Channel and (iv) Enzymes proposed by Yamanishi et al. [25], which were constructed from the intersection of the databases KEGG Brite, BRENDA, and DrugBank, in addition to (v) Global proposed by Wu et al. [32], which was obtained from the DrugBank database. For external time-split validation we derived a new dataset from ChEMBL dubbed ChEMBL24→28 [37,38] (Table 2). First, we prepared two separate datasets from ChEMBL versions 24 (released in June 2018) and 28 (released in February 2021). Briefly, DTIs with K ${}_{i}$/K ${}_{d}$/IC ${}_{50}$/EC ${}_{50}$ < 10 µM of drugs or clinical candidates (clinical phase ≥ 1) with 5 < #atoms < 80 to human proteins were included (see the Appendix A for more details). Then, we filtered both datasets to only include interactions of targets present in both sets and subsequently removed all DTIs from the ChEMBL28 set that were already included in the ChEMBL24 set. Finally, we defined the ChEMBL24-derived dataset as the training set and the ChEMBL28-derived set as the test set. Consequently, the training and test set shared the same targets, but their ligands were mutually exclusive. Test set DTIs were of newer date than training set DTIs (time split).

### 3.2. Molecular Descriptors and Similarity Measures

MACCS keys (MACCS) and extended-connectivity fingerprint with radius equal to 2 (ECFP4) were calculated with the software OpenBabel version 3.1.0 [41]; functional extended-connectivity fingerprint with radius equal to 2 (FCFP4) was calculated using the software RDKit version 2021_09_5 [42]; Klethora–Roth fingerprint (KR) were calculated with the PaDEL-Descriptor software [43]. For this group of fingerprints, the Tanimoto coefficient was calculated as the measure of similarity [36]. The Tanimoto coefficient was selected based on its general robust performance in fingerprint-based similarity calculations [44]. As an alternative, we tested the Tversky index (with α = 0.0 and β = 1.0) on our external time-split dataset and obtained inferior results (Appendix A). As a proof of concept, we also calculated the real-valued molecular descriptor Mold ${}^{2}$ [34], constructed from a diverse set of 777 descriptors calculated from the molecular formula (one-dimensional descriptors) and two-dimensional molecular graph (two-dimensional descriptors). These descriptors can be divided into 20 groups ranging from counts (e.g., number of atoms and bonds) over continuous physicochemical properties (e.g., molecular weight, logP) to topological indices (e.g., Wiener and Zagreb index). Mold ${}^{2}$ (version 2.0) was designed for cheminformatics and toxicoinformatics problems and is available online free of charge from [45]. This descriptor was normalized column-wise on [0,1] prior to applying the continuous formula of the Tanimoto coefficient [36]. The datasets are available from SimSpread’s GitHub repository [46].

### 3.3. Chemical Similarity Feature Matrix Construction

A similarity matrix S can be generated by calculation of a similarity metric between all pairs of descriptor vectors of the ligands of a DTI dataset (Figure 1A). From S we can construct a similarity-based feature matrix $S^{\prime }$ by applying a similarity threshold α:(1)$$S_{ij}^{\prime }=\left\{\begin{array}{cc} w(i,j) & \mathrm{if}\;S_{ij}\geq \alpha ;\\ 0 & \mathrm{otherwise}. \end{array}\right.$$
where S corresponds to the chemical similarity matrix, $S^{\prime }$ to the final feature matrix, *i* and *j* to compounds in the studied dataset, and w(i,j) the weighting scheme employed for feature matrix construction.

We propose two distinct weighting schemes: (i) a binary weighting scheme (denoted as $S_{bin}^{\prime }$) and (ii) a chemical similarity weighting scheme (denoted as $S_{sim}^{\prime }$). In binary weighting, the compound feature weight is either 1, if $S_{ij}$ is greater than or equal to α, or 0, otherwise. In contrast, in the chemical similarity weighting scheme, the weight is $S_{ij}$, if $S_{ij}$ is greater than or equal to α, or 0, otherwise. In the former scheme, both distant and close chemical relationships are equally weighted, while in the latter scheme, the higher the similarity between two compounds, the higher the importance of the corresponding feature in $S^{\prime }$.

### 3.4. *De Novo* Network-Based Inference Method

Wu et al. (2017) proposed a generalized variant of NBI to enable de novo prediction of drug–target interactions using a trilayered network constructed from molecular substructures, drugs, and biological targets [32]. Here, we propose a further generalization of this formalism by substituting the aforementioned substructure layer with a generalized feature layer. In this feature layer, small molecules are represented by non-negative real-valued feature vectors.

Denoting a set of $N_{F}$ features as $F=\{f_{1},f_{2},\ldots ,f_{N_{F-1}},f_{N_{F}}\}$, a set of $N_{C}$ novel compounds without known targets as $C=\{c_{1},c_{2},\ldots ,c_{N_{C-1}},c_{N_{C}}\}$, a set of $N_{D}$ drugs with known targets as $D=\{d_{1},d_{2},\ldots ,d_{N_{D-1}},d_{N_{D}}\}$, and a set of $N_{T}$ biological targets as $T=\{t_{1},t_{2},\ldots ,t_{N_{T-1}},t_{N_{T}}\}$, a compound–feature–drug–target graph can be represented as the graph G(V,E), where V=C∪F∪D∪T is the node set and $E=E_{FC}\cup E_{FD}\cup E_{DT}$ is the edge set constructed from the edges between features and novel compounds, between features and drugs and between drugs and targets, respectively.

A resource diffusion algorithm is applied for target prediction to the $V^{\prime }=F\cup D\cup T$ subgraph (Figure 1C). Briefly, all nodes are initialized with a resource value of zero. Then, for each novel compound in C, initial resources are assigned to the first network layer corresponding to the values of F (Figure 1D, upper panel). These resources are then distributed to the second layer D by first dividing the resource value of each node in F by the node’s degree and then adding the resulting fraction to the resource value of all linked nodes in D (Figure 1D, middle panel). Next, a second round of resource spreading is applied, which distributes the resources from D back to F and forward to T by applying the same division & summation procedure (Figure 1D, lower panel). The target prediction scores correspond to the final resources of the third layer T. The higher the resource value of a target, the higher the prediction probability of this compound-target interaction. This resource diffusion algorithm is equivalent to Wu et al. (2017) [32] and the mathematical details are explained in the Appendix A. The algorithm is based on several matrix multiplications, which we readily parallelized on modern video graphics hardware (GPU) for higher efficiency.

### 3.5. SimSpread Algorithm

SimSpread employs a tripartite drug–drug–target network where resources are distributed based on the NBI formalism, and small molecules are described by their similarity to other molecules. The approach is outlined as follows:

*Step 1*: Two chemical libraries need to be prepared: (i) a screening library (novel compounds without known targets or compounds for drug repurposing) and (ii) a drug–target interaction library, e.g., ChEMBL (Figure 1B). Then, two chemical similarity feature matrices of i) novel compounds vs. drugs and ii) drugs vs. drugs need to be generated (Figure 1A). For this, an arbitrary molecular descriptor may be calculated for the molecules, followed by the calculation of a similarity matrix S by applying an appropriate similarity metric (Tanimoto coefficient in our case). Next, S is transformed into a feature matrix $S^{\prime }$ by applying a similarity threshold α and an appropriate weighting scheme, as described above (Equation (Equation 1)).

*Step 2*: A compound–feature–drug–target network is created, where edges between compounds and features are given by non-zero values of the compound–drug feature matrix, edges between features and drugs are given by non-zero values of the drug–drug matrix, and edges between drugs and targets are based on known target annotations (Figure 1C). It should be noted that in SimSpread, feature nodes correspond to drugs. Hence, the node sets F and D are of the same length and correspond to the same drugs.

*Step 3*: De novo network-based inference is applied to spread resources across the network as explained in the previous section (Figure 1D). The amount of resources that reach a target corresponds to the probability of a compound interacting with the given target, where more resources correspond to higher probabilities.

The proposed method was implemented in Julia (version 1.7.3) using the Base and Standard libraries in conjunction with the CUDA.jl package for GPU acceleration. Computing times for 10-times 10-fold cross-validation on the five benchmarks datasets on GPU-accelerated hardware ranged from 7 to 140 s (Appendix A). Source code is available from SimSpread’s GitHub repository [46].

### 3.6. Other Predictive Algorithms

SimSpread was compared with other predictive methods based on (i) chemical similarity, in the form of *k*-NN, and (ii) network-based inference, in the form of NBI and SDTNBI. For *k*-NN (*k*-nearest neighbors), an in-house implementation of the algorithm was devised using matrix multiplication for GPU parallelization. This method compares a query compound with annotated ligands of drug targets and returns the highest similarity (*k* = 1) as the prediction value for each target. For NBI, we reimplemented the method based on matrix multiplication as published in [47]. This method employs a bipartite drug–target network to predict interactions between both sets of nodes based on a resource-spreading algorithm derived from social network studies. For SDTNBI, we reimplemented the method based on matrix multiplication as described in [32]. This method employs a trilayered substructure–drug–target network to predict drug–target interactions through the NBI algorithm. The methods mentioned here were reimplemented in Julia (1.7.3) using the Base and Standard libraries in conjunction with the CUDA.jl package for GPU acceleration. Validation analyses for the reimplementation of these methods are available in the Appendix A. Source code is available from SimSpread’s GitHub repository [46].

### 3.7. Cross-Validation and Time-Split Validation

We employed leave-one-out (LOO) and 10-times 10-fold cross-validation (CV) for parameter optimization and benchmarking [37,48]. Splitting into test and training data was performed over drugs, i.e., if a drug formed part of the test set, all of its edges to targets (DTIs) were removed. If a drug formed part of the training set, DTIs were conserved and used to predict the interactions of the test set. Due to the nature of this data splitting, some targets may become disconnected from the network during CV and, therefore, can never be correctly predicted. These cases were ignored when calculating evaluation metrics.

For time-split validation, the ChEMBL24→28 dataset was split into a training set consisting of DTIs present in ChEMBL24 and a test set consisting of all newer DTIs found in ChEMBL28 [38]. The prediction network was generated from the training set, while target predictions were performed for all compounds of the test set (ignoring their target annotations). The predictions were evaluated by comparing the predicted DTIs with the known DTIs of the test set.

For LOO CV, the evaluation metrics were calculated per drug, for 10-times 10-fold CV per fold, and in time-split validation per drug. The resulting distributions of metric values were reported in terms of their central tendency (average, median) and dispersion (standard deviation, interquartile range).

### 3.8. Evaluation Metrics

We employed several evaluation metrics that can be classified into three groups: (i) overall performance, (ii) early recognition, and (iii) binary prediction performance. The area under the Receiver Operating Characteristic curve (AuROC) and area under the Precision–Recall curve (AuPRC) were calculated to test the overall performance. Early recognition was assed by Precision of the top-20 predictions (P@20), Recall of the top-20 predictions (R@20) [32], and by Boltzmann-Enhanced Discrimination of ROC (AuBEDROC) with α = 20 [48]. Finally, binary predictive performance was assessed by calculating the maximum scores of the Matthews correlation coefficient (MCC), F ${}_{1}$ score, and Balanced Accuracy (bACC). Here, we calculated the score for each metric at all possible discrimination thresholds and then retrieved the maximum observed value. Please refer to the Appendix A for mathematical details.

### 3.9. Exploration of Chemical and Biological Space

The diversity and novelty of DTI predictions were analyzed for drugs (chemical space) and targets (biological space). The best 250 time-split predictions were selected, and their unique drugs and targets were extracted. Murcko scaffolds were calculated using RDKit (version 2021_09_5) [42] for the compounds, and their ECFP4 descriptors were calculated. We classified scaffolds as “known” if they were present in both the test and training set and “novel” if they were only found in the test set. Scaffold descriptors were projected on a two-dimensional (2D) space with the help of the UMAP algorithm (version 0.5.3), a flexible non-linear dimension reduction algorithm based on manifold learning [40]. A higher number of unique scaffolds (known + novel) and larger coverage of the map area were associated with higher chemical diversity. A higher number of novel scaffolds was associated with better scaffold hopping capabilities.

Targets were described by constructing a binary (presence/absence) vector of gene ontology (GO) terms (molecular function, biological process, and cellular component) obtained from the UniProt database for each of the targets [49]. Since all targets were shared between the time-split training and test sets, known/novel classification was omitted. Finally, 2D maps were generated with UMAP. A higher number of unique targets and larger coverage of the map area were associated with higher target diversity.

To analyze the effect of selecting different numbers of top *L* considered results, we varied *L* from 10 to 1000 in steps of 10. We analyzed the number of unique scaffolds and targets as a function of *L*.

## 4. Conclusions

We proposed SimSpread, a novel method to predict the interactions between small molecule compounds and biological targets guided by chemical similarity as a meta-description of small molecules and network-based inference for target prediction. The proposed method consists of three main steps: (i) construction of chemical similarity matrices of screening compounds vs. drugs and drugs vs. drugs, (ii) transformation of the similarity matrices into feature matrices by applying a similarity cutoff and weighting, and (iii) target prediction with the help of de novo network-based inference in a three-layered drug–drug–target network. We optimized free method parameters (choice of molecular descriptor, similarity cutoff, weighting scheme) to obtain the best performance. Our benchmark results suggest that SimSpread is competitive with other network-based (SDTNBI) and chemical similarity-based (*k*-NN) methods and, in many cases, surpassed their prediction performance. Finally, we showcased that SimSpread can achieve a more balanced exploration of chemical ligand space and biological target space than other methods, enabling scaffold and target hopping.

As an example, we generated target predictions for nintedanib and pirfenidone, two drugs with antifibrotic properties, which are used to treat idiopathic pulmonary fibrosis [50,51]. Predictions were generated with SimSpread ${}_{sim}$ against our complete ChEMBL28 dataset (1815 drugs & clinical candidates; 1069 targets), and neither drug was included in the prediction network. Nintedanib is a tyrosine kinase inhibitor, and several of its known pharmacological targets were identified by SimSpread (Appendix A): Vascular endothelial growth factor receptor (VEGFR-1, -2, and -3), fibroblast growth factor receptor (FGFR1, -2 and -3), platelet-derived growth factor receptor (PDGFR-α and -β), receptor tyrosine kinase FLT-3 and non-receptor tyrosine kinases Lck, Lyn, and Src [50]. The precise molecular targets of pirfenidone are not known [51], and we include the top-15 predictions for interest (Appendix A).

A primary advantage of SimSpread is that small molecules are represented by feature vectors of similarity indices providing a flexible means of employing diverse molecular descriptions. Molecular descriptors are not limited to two-dimensional fingerprints derived from two-dimensional structures of compounds. Here, we included the real-valued topological descriptor Mold ${}^{2}$ in addition to traditional binary fingerprints. However, other molecular representations such as three-dimensional compound shape, structural similarity of protein binding sites, and three-dimensional pharmacophore descriptors are compatible with SimSpread. Exploring them will be interesting in the future. SimSpread performed well on small datasets, e.g., Nuclear Receptors (54 ligands, 26 targets), and prediction performance did not depend on target degree (number of annotated ligands per target). On larger datasets, e.g., Global (1844 ligands, 1032 targets), SimSpread outperformed other methods. An upper limit for dataset size is only given by the available computational resources. A limitation of SimSpread is that it is impossible to generate predictions for query compounds whose similarity to every compound of the first network layer is below the threshold α. In this case, the length of the feature vector is zero, resource spreading is not possible, and the predicted value is zero for all targets. However, this case was rare for our five benchmark datasets and occurred for 4.7% of all ligands. Interestingly, this characteristic of SimSpread can be seen as an intrinsic notion of its application domain. No target predictions are generated for query ligands outside SimSpread’s application domain instead of returning likely meaningless targets.

We believe that SimSpread will be useful for predicting biological targets for drug repositioning and polypharmacology, predicting off-target effects, and virtual screening of novel small molecule compounds in drug discovery.

## Figures and Tables

**Figure 1 ijms-23-09666-f001:**
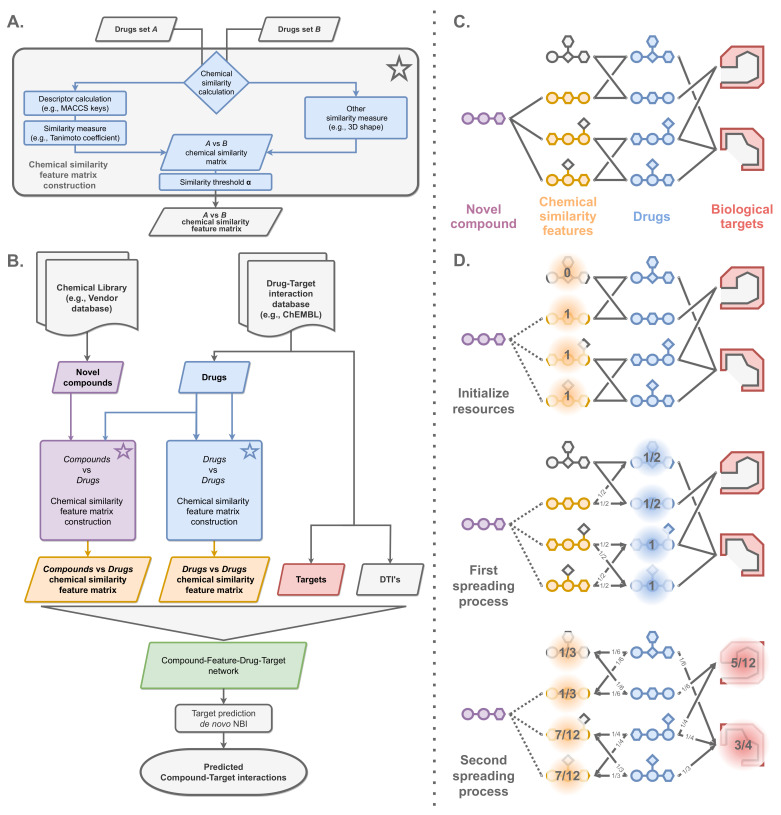
**Schematic representation of SimSpread.** (**A**) Chemical similarity feature matrix construction from two different sets of drugs. (**B**) De novo target prediction pipeline of SimSpread. (**C**) Tetra-layered network employed by SimSpread for de novo target prediction. (**D**) Schematic representation of the de novo NBI resource-spreading algorithm on the tri-layered drug–drug–target subnet. Here, fractions placed on the nodes represent the amount of resources stored in a given resource-spreading process, arrows indicate resource spreading direction and fractions in arrows represent the amount of resources distributed. Please refer to the Section 3 for more information.

**Figure 2 ijms-23-09666-f002:**
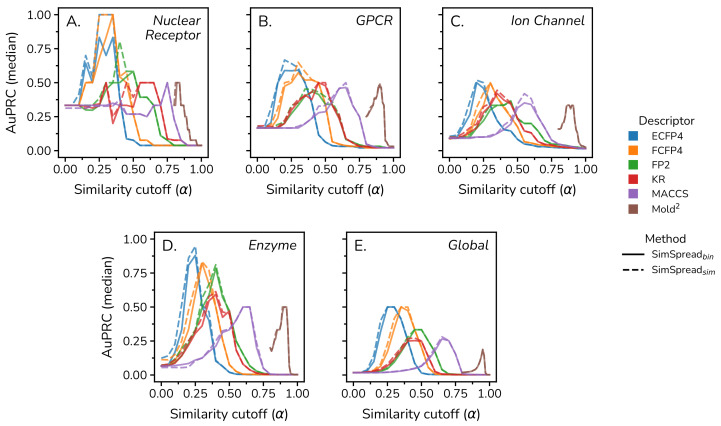
**SimSpread parameter optimization.** Median AuPRC obtained for LOOCV over five datasets for different α values. Datasets: (**A**) Nuclear Receptor, (**B**) GPCR, (**C**) Ion Channel, (**D**) Enzyme, and (**E**) Global. A total of six different molecular descriptors were tested for method optimization: ECFP4 (blue), FCFP4 (orange), FP2 (green), KR (red), MACCS kys (purple) and Mold ${}^{2}$ (brown). Solid lines correspond to SimSpread ${}_{bin}$ while dashed lines correspond to SimSpread ${}_{sim}$. Charts for other performance metrics and cross-validation schemes are available from the Appendix A.

**Figure 3 ijms-23-09666-f003:**
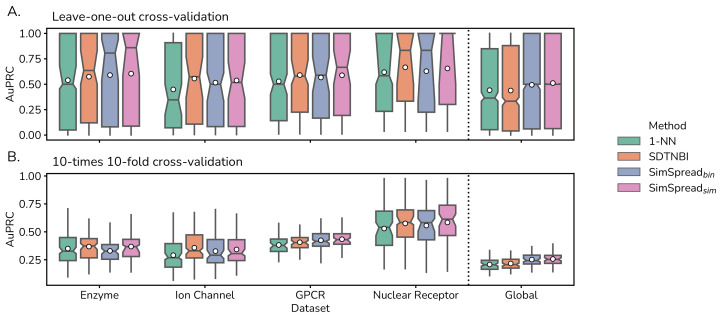
**Predictive performance comparison with other methods in LOO and 10-fold cross-validation.** Boxplots for AuPRC obtained from (**A**) leave-one-out and (**B**) 10-times 10-fold cross-validation over five datasets for 1-NN, SDTNBI, SimSpread ${}_{bin}$ and SimSpread ${}_{sim}$ using the ECFP4 molecular descriptor and Tanimoto coefficient as similarity measure. For SimSpread variants, the similarity cutoff corresponds to the one that maximizes mean AuPRC per dataset. Boxplot notch represents the bootstrapped 95% confidence interval (CI) around the median, the white dot represents the mean and whiskers correspond to 1.5× interquartile range (IQR). Charts for other performance metrics are available from the Appendix A.

**Figure 4 ijms-23-09666-f004:**
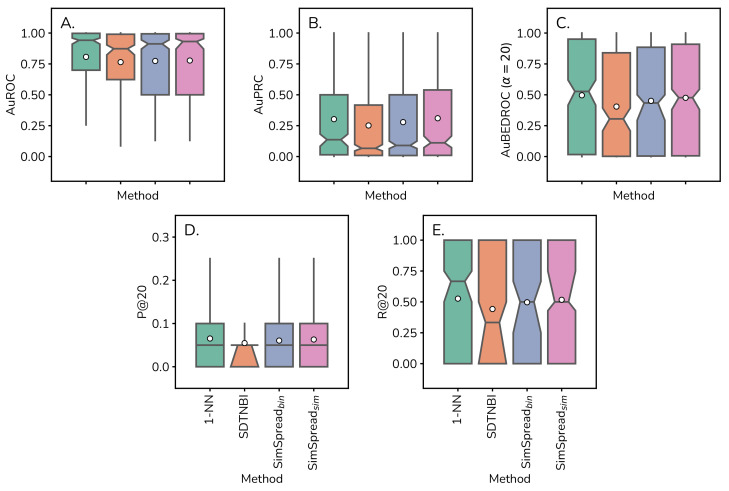
**Predictive performance comparison of four methods in time-split validation.** Boxplots for (**A**) AuROC, (**B**) AuPRC, (**C**) AuBEDROC, (**D**) P@20, and **(E)** R@20 obtained in the time-split validation on the ChEMBL24→28 dataset for 1-NN, NBI, SDTNBI, SimSpread ${}_{bin}$ and SimSpread ${}_{sim}$ using ECFP4 molecular descriptor and Tanimoto coefficient as similarity measure. For SimSpread variants, the similarity cutoff α is equal to 0.2. Boxplot notch represents the bootstrapped 95% CI around the median, the white dot represents the mean, and whiskers correspond to 1.5× IQR.

**Figure 5 ijms-23-09666-f005:**
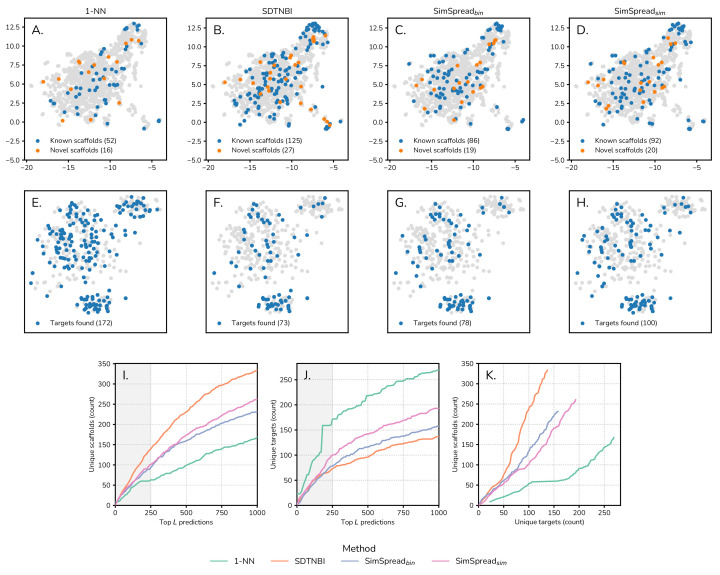
**Chemical and biological space exploration.** (**A**–**D**) Molecular scaffolds and (**E**–**H**) biological targets found in the best 250 predictions obtained in the time-split validation over the ChEMBL24→28 dataset using ECFP4 molecular descriptor for 1-NN, SDTNBI, SimSpread ${}_{bin}$ and SimSpread ${}_{sim}$. (**I**–**J**) Quantification and (**K**) relationship of the number of unique scaffolds and unique targets found in the best 1000 predictions for the previously mentioned methods and SimSpread ${}_{bin}$. The shadowed areas in panels **I** and **J** correspond to the predictions shown in panels **A** through **D** and **E** through **H**, respectively. The 2D projection of the chemical space was calculated with UMAP over the ECFP4 molecular descriptors of Murcko scaffolds and the 2D projection of the target space was obtained with UMAP over the GO terms of the biological targets.

**Table 1 ijms-23-09666-t001:** Predictive performance comparison of the studied methods in time-split validation.

Method	AuROC	AuPRC	AuBEDROC	P@20	R@20
1-NN	**0.942 (0.699, 0.997)**	**0.137 (0.014, 0.500)**	**0.527 (0.016, 0.951)**	**0.050 (0.000, 0.100)**	**0.667 (0.000, 1.000)**
SDTNBI	0.873 (0.623, 0.990)	0.067 (0.009, 0.417)	0.305 (0.002, 0.840)	*0.050 (0.000, 0.050)*	0.333 (0.000, 1.000)
SimSpread ${}_{bin}$	0.912 (0.500, 0.994)	0.090 (0.009, 0.500)	0.434 (0.004, 0.885)	**0.050 (0.000, 0.100)**	*0.500 (0.000, 1.000)*
SimSpread ${}_{sim}$	*0.932 (0.500, 0.995)*	*0.111 (0.010, 0.539)*	*0.477 (0.006, 0.909)*	**0.050 (0.000, 0.100)**	*0.500 (0.000, 1.000)*

Note: Fingerprint used is ECFP4 with α = 0.2. Each value corresponds to the median with IQR in parenthesis obtained for general and early-recognition evaluation metrics used. Bold typeface corresponds to the best median performance and italic typeface corresponds to the second best median performance.

**Table 2 ijms-23-09666-t002:** Description of datasets.

	Statistics						
**Dataset**	$N_{\mathrm{Ligands}}$	$N_{\mathrm{Targets}}$	$N_{\mathrm{Interactions}}$	$k_{\mathrm{Ligands}}$	$k_{\mathrm{Targets}}$	$N_{\mathrm{Atoms}}$	**Density**
Enzyme	445	664	2.926	6.57 ± 12.74	4.41 ± 6.33	[7, 101]	0.99
Ion Channel	223	95	635	7.03 ± 12.14	7.24 ± 6.63	[7, 95]	3.45
GPCR	210	205	1.476	2.85 ± 3.35	6.68 ± 8.43	[7, 94]	3.00
Nuclear Receptor	54	26	90	1.67 ± 1.60	3.46 ± 3.39	[14, 59]	6.41
Global	1844	1032	10,185	5.52 ± 15.79	9.87 ± 14.95	[5, 118]	0.54
ChEMBL24→28	1513	395	4863	3.21 ± 5.66	12.31 ± 12.24	[7, 78]	0.81

Note: N = Amount; k = Degree; N_Atoms_ = Range of the number of heavy atoms (denoted as [min N_Atoms_, max N_Atoms_,]); Density = Ratio between the number of annotated and all possible interactions between ligands and targets.

## Data Availability

Source code of the SimSpread method, datasets and benchmarking/analysis is available from a public GitHub repository [46].

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
