# Peer review of "De Novo Prediction of Drug Targets and Candidates by Chemical Similarity-Guided Network-Based Inference"

_ijms, 2022, doi:10.3390/ijms23179666_

Round 1
Reviewer 1 Report
Interesting chemogenomics study. I am not an expert in that field, but the authors managed to spark my curiosity. The manuscript is already publication quality and the study is pretty thorough; it changes from what I am usually asked to review.
I only have minor remarks:
- in the introduction, it would be nice to give an overview of the Mold^2 [33] molecular descriptor, rather than just pointing to the corresponding paper
- In Figure 3, I would like to see AuROC and PR@20, so that we can compare with Figure 4
- For future research, maybe try to encode only the sequence of the binding site of proteins (rather than encoding whole proteins); it might increase the performance. If you don't have structural data (protein-ligand complexes) to find the binding site; try to keep only the most conserved part of each sequence within its protein family (maybe that would introduce one more parameter, but really worth trying)
- the performance analysis is only global: i) can you show us the top-20 predicted by your method protein targets for some known drugs (like Nintedanib, Pirfenidone and Clofoctol). If your protein dataset has some viral proteins, that might be interesting for Clofoctol (its covid-19 antiviral activity has strong experimental evidence in-vivo). ii) For the reverse task, can you suggest a handful of interesting drug-repurposing examples derived by your method. For the three same molecules, that would be interesting.
- how is the performance of your method evolving versus the number of known ligands for a given protein target? I'd like to have an idea of how many ligands need to be known for a target so that it starts to be useful for such chemogenomics methods.
Author Response
Response to reviewer 1:
We thank the reviewer for his/her useful comments and kind suggestions. We provide a point by point response below and hope to have addressed all points raised to the satisfaction of the reviewer. We also used the English language editing service Grammarly to improve the writing style.
Comment 1:
- in the introduction, it would be nice to give an overview of the Mold^2 [33] molecular descriptor, rather than just pointing to the corresponding paper
Response to comment 1:
We modified the respective text in the Results section to provide an overview of the Mold2 molecular descriptor: “We included the real-valued Mold2 descriptor in our analyses, which consists of a diverse set of 777 individual one-dimensional and two-dimensional molecular descriptors, such as atoms counts, physicochemical properties and topological features, calculated from the two-dimensional chemical structure [33].”
We further extended the description in the methods section: “As a proof of concept, we also calculated the real-valued molecular descriptor Mold2 [33], constructed from a diverse set of 777 descriptors calculated from the molecular formula (one-dimensional descriptors) and two-dimensional molecular graph (two-dimensional descriptors). These descriptors can be divided into 20 groups ranging from counts (e.g. number of atoms and bonds), over continuous physicochemical properties (e.g. molecular weight, logP), to topological indices (e.g. Wiener and Zagreb index). Mold2 (version 2.0) was designed for cheminformatics and toxicoinformatics problems and is available online free of charge from [44].”
[33] Hong, H.; Xie, Q.; Ge, W.; Qian, F.; Fang, H.; Shi, L.; Su, Z.; Perkins, R.; Tong, W. Mold2 , Molecular Descriptors from 2D Structures for Chemoinformatics and Toxicoinformatics. Journal of Chemical Information and Modeling 2008, 48, 1337–1344. https://doi.org/10.1021/ci800038f.
[44] Mold2 | FDA. https://www.fda.gov/science-research/bioinformatics-tools/mold2. Accessed 27 of July, 2022.
Comment 2:
In Figure 3, I would like to see AuROC and PR@20, so that we can compare with Figure 4
Response to comment 2:
AuROC and PR@20 (and all the other performance metrics) for the analysis presented in Fig. 3 are available from the supplementary material. We preferred not to add this information to Fig. 3 for clarity reasons. We added the following sentence to the caption of Fig. 3: “Charts for other performance metrics are available from the supplementary material Figures S3 and S5.”
Comment 3:
For future research, maybe try to encode only the sequence of the binding site of proteins (rather than encoding whole proteins); it might increase the performance. If you don't have structural data (protein-ligand complexes) to find the binding site; try to keep only the most conserved part of each sequence within its protein family (maybe that would introduce one more parameter, but really worth trying)
Response to comment 3:
We greatly appreciate the Reviewer’s suggestion to incorporate binding site information into SimSpread. In fact, we are currently extending SimSpread to consider the structural similarity of three-dimensional protein binding sites and so far obtained promising preliminary results.
Comment 4:
The performance analysis is only global: i) can you show us the top-20 predicted by your method protein targets for some known drugs (like Nintedanib, Pirfenidone and Clofoctol). If your protein dataset has some viral proteins, that might be interesting for Clofoctol (its covid-19 antiviral activity has strong experimental evidence in-vivo). ii) For the reverse task, can you suggest a handful of interesting drug-repurposing examples derived by your method. For the three same molecules, that would be interesting.
Response to comment 4:
We did not find any of the three compounds in our datasets, so we derived a prediction network from the complete ChEMBL28 set and ran predictions for nintedanib and pirfenidone. We excluded clofoctol (an antibiotic) from this analysis, as our ChEMBL28 set only includes human targets. We included the top-15 target predictions for nintedanib and pirfenidone in the supplementary material. Several of nintedanib (a kinase inhibitor) known targets were identified. We added the following phrase to the conclusions: “As an example we generated target predictions for nintedanib and pirfenidone, two drugs with antifibrotic properties, which are used to treat idiopathic pulmonary fibrosis [48, 49]. Predictions were generated with SimSpreadsim against our complete ChEMBL28 dataset (1815 drugs & clinical candidates; 1069 targets) and neither drug was included in the prediction network. Nintedanib is a tyrosine kinase inhibitor and several of its known pharmacological targets were identified by SimSpread (Table S22): Vascular endothelial growth factor receptor (VEGFR-1, -2 and -3), fibroblast growth factor receptor (FGFR1, -2 and -3), platelet-derived growth factor receptor (PDGFR-α and -β), receptor tyrosine kinase FLT-3 and non-receptor tyrosine kinases Lck, Lyn and Src [48]. The precise molecular targets of pirfenidone are not known [49] and we include the top-15 predictions for interest (Table S22).”
[48] Lamb YN. Nintedanib: A Review in Fibrotic Interstitial Lung Diseases. Drugs. 2021 Apr;81(5):575-586. doi: 10.1007/s40265-021-01487-0. Epub 2021 Mar 25. Erratum in: Drugs. 2021 Apr 13;: Erratum in: Drugs. 2021 Jun;81(9):1133. PMID: 33765296; PMCID: PMC8163683.
[49] Aimo A, Spitaleri G, Nieri D, Tavanti LM, Meschi C, Panichella G, Lupón J, Pistelli F, Carrozzi L, Bayes-Genis A, Emdin M. Pirfenidone for Idiopathic Pulmonary Fibrosis and Beyond. Card Fail Rev. 2022 Apr 14;8:e12. doi: 10.15420/cfr.2021.30. PMID: 35516794; PMCID: PMC9062707.
Comment 5:
How is the performance of your method evolving versus the number of known ligands for a given protein target? I'd like to have an idea of how many ligands need to be known for a target so that it starts to be useful for such chemogenomics methods.
Response to comment 5:
We thank the reviewer for this interesting question. We added an additional analysis to the supplementary material (Figure S7) with scatter plots of performance metrics vs. the target degree, i.e., the number of annotated ligands per target. We added the following paragraph to section 2.3:
“We were interested in a possible dependence of the prediction performance on the target degree, i.e the number of annotated ligands per target. We performed linear regression analyses of all performance metrics of the time-split validation vs. target degree (Figure S7). Of eight performance metrics analyzed, only three depended significantly on target degree (p < 0.05, Wald test with t distribution). Effect sizes (coefficients of determination) were low (0.00004 < R^2 < 0.1). We conclude that in our time-split dataset performance did not generally depend on the number of annotated ligands per target. We believe that our network-based SimSpread method is particularly suited for prediction in datasets with low target degrees.”
Reviewer 2 Report
The authors presented an interesting and useful approach SimSpread, which combines network-based inference with chemical similarity, and can be used for de novo prediction of drug targets and candidates,
I have the following comments:
The five benchmark datasets, which the authors used and which were previously employed in similar works should be available in supplementary data, to make the results of the publication reproducible. Also, the external time-split dataset derived from ChEMBL should be available in the supplementary data. The authors mentioned, that it is somewhere in Github, but I was not able to find, where is for example the Ion Channel dataset.
Authors should explain, why they selected the Tanimoto coefficient. Did they also try other similarity coefficients?
Which limitations has SimSpread approach? E.g., is it limited to 2D structures of compounds and 2D descriptors? How large should be a training set, which can be used for SimSpread? In which cases SimSpread can not be applied? The authors should mention this in the article.
Why does Figure 5 miss SimSpread_bin?
Table 2 should also include information about the sizes of ligands and targets (intervals from and to).
How (by which tool or script) do the authors calculate descriptor Mold2?
Were the reimplementations of k-NN and NBI tested on original data, and used in publications of these methods? And provided the reimplementations with the same results, which are presented in the original publications?
The article missed information about the time complexity and duration of the SimSpread algorithm.
Author Response
Response to Reviewer 2:
We thank the reviewer for his/her useful comments and kind suggestions. We provide a point by point response below and hope to have addressed all points raised to the satisfaction of the reviewer. We also used the English language editing service Grammarly to improve the writing style.
Comment 1:
The five benchmark datasets, which the authors used and which were previously employed in similar works should be available in supplementary data, to make the results of the publication reproducible. Also, the external time-split dataset derived from ChEMBL should be available in the supplementary data. The authors mentioned, that it is somewhere in Github, but I was not able to find, where is for example the Ion Channel dataset.
Response to comment 1:
We appreciate this comment and apologize for the lack of clarity. We added readme files to the github repository to facilitate easier access to the datasets. For the Yamanishi (2008), Wu (2017) and our time-split dataset we provide the chemical compound structures in SMILES format, drug-drug similarity matrices, precalculated molecular descriptor tables and drug-target interaction matrices. Further, for our time-split dataset we provide an additional table with detailed information about the drug-target interactions, e.g. bioactivity values. The datasets are available from: https://github.com/cvigilv/simspread/tree/main/data. We added the following sentence to the Materials and Methods section: “The datasets are available from GitHub: https://github.com/cvigilv/simspread.”
Comment 2:
Authors should explain, why they selected the Tanimoto coefficient. Did they also try other similarity coefficients?
Response to comment 2:
We consider that the Tanimoto coefficient is a widely employed similarity metric that provides robust performance in many scenarios [43]. However, we tried other similarity indices, such as the Tversky index, and their performance was generally inferior to the Tanimoto coefficient. We added this analysis to the supplementary material (Figure S8). We further added the following text to section 3.2. Molecular descriptors and similarity measures: “The Tanimoto coefficient was selected based on its general robust performance in fingerprint-based similarity calculations [43]. As an alternative, we tested the Tversky index (with α = 0.0 and β = 1.0) on our external time-split dataset and obtained inferior results (Figure S8).”
[43] Bajusz D, Rácz A, Héberger K. Why is Tanimoto index an appropriate choice for fingerprint-based similarity calculations? J Cheminform. 2015 May 20;7:20. doi: 10.1186/s13321-015-0069-3.
Comment 3:
Which limitations has SimSpread approach? E.g., is it limited to 2D structures of compounds and 2D descriptors? How large should be a training set, which can be used for SimSpread? In which cases SimSpread can not be applied? The authors should mention this in the article.
Response to comment 3:
We thank the reviewer for pointing out the missing discussion of advantages and limitations of SimSpread. We added the following paragraph to the Conclusions section:
“A main advantage of SimSpread is that small molecules are represented by feature vectors of similarity indices providing a flexible means of employing diverse molecular descriptions. Molecular descriptors are not limited to two-dimensional fingerprints derived from two-dimensional structures of compounds. Here, we included the real-valued topological descriptor Mold2 in addition to traditional binary fingerprints. However, other molecular representations such as three-dimensional compound shape, structural similarity of protein binding sites, and three-dimensional pharmacophore descriptors are compatible with SimSpread. Exploring them will be interesting in the future. SimSpread performed well on small datasets, e.g. Nuclear Receptors (54 ligands, 26 targets) and prediction performance did not depend on target degree (number of annotated ligands per target). On larger datasets, e.g. Global (1,844 ligands, 1,032 targets) SimSpread outperformed other methods. An upper limit for dataset size is only given by the available computational resources. A limitation of SimSpread is that it is impossible to generate predictions for query compounds whose similarity to every compound of the first network layer is below the threshold alpha. In this case the length of the feature vector is zero, resource spreading is not possible and the predicted value is zero for all targets. However, for out five benchmark datasets this case was rare and occurred for 4.7% of all ligands. Interestingly, this characteristic of SimSpread can be seen as an intrinsic notion of its application domain. No target predictions are generated for query ligands outside SimSpread’s application domain, instead of returning likely meaningless targets.”
Comment 4:
Why does Figure 5 miss SimSpread_bin?
Response to comment 4:
We appreciate this comment from the reviewer. We originally excluded SimSpread_bin to improve clarity, however, we agree with the reviewer that this decision was inconsistent with the rest of the paper. We added SimSpread_bin to Figure 5 and adjusted UMAP parameters for a better visualization of all four methods. We also added a discussion of SimSpread_bin to the main text (seccion 2.4) and it behaved similarly to SimSpread_sim.
Comment 5:
Table 2 should also include information about the sizes of ligands and targets (intervals from and to).
Response to comment 5:
We included the range of the number of heavy atoms (min-max) for all datasets in Table 2. These ranges are similar for all datasets analyzed.
Comment 6:
How (by which tool or script) do the authors calculate descriptor Mold2?
Response to comment 6:
We added the following phrase to section 3.2. Molecular descriptors and similarity measures: “Mold2 (version 2.0) was designed for cheminformatics and toxicoinformatics problems and is available online free of charge from [44].”
[44] Mold2 | FDA. https://www.fda.gov/science-research/bioinformatics-tools/mold2. Accessed 27 of July, 2022.
Comment 7:
Were the reimplementations of k-NN and NBI tested on original data, and used in publications of these methods? And provided the reimplementations with the same results, which are presented in the original publications?
Response to comment 7:
We appreciate this comment and added validations to the supplementary material. We validated the reimplementation of SDTNBI against the published values in Wu, et al (2016) on the Global dataset with minimal deviation (Table S20). We validated our network-based k-NN against an in-house implementation of k-NN written in C++ and based on Gfeller et al (2013), also with minimal deviation (Table S21). We added the following sentence to the Materials and Methods section: “Validation analyses for the reimplementation of these methods are available in the supplementary material (Tables S20 and S21).”
Wu, Z.; Cheng, F.; Li, J.; Li, W.; Liu, G.; Tang, Y. SDTNBI: an integrated network and chemoinformatics tool for systematic prediction of drug–target interactions and drug repositioning. Briefings in Bioinformatics 2016, p. Bbw012.
Gfeller et al (2013). Shaping the interaction landscape of bioactive molecules. Bioinformatics, 29(23), 3073-3079. https//doi.org/10.1093/bioinformatics/btt540.
Comment 8:
The article missed information about the time complexity and duration of the SimSpread algorithm.
Response to comment 8:
We added Table S19 to the supplementary material with computation times of SimSpread_sim in 10-times 10-fold cross-validation for the 5 benchmark datasets on GPU-accelerated hardware. We added the following text to section 3.5. SimSpread algorithm: “Computing times for 10-times 10-fold cross-validation on the five benchmarks datasets on GPU-accelerated hardware ranged from 7 to 140 seconds (Table S19).”